# Cell-free DNA screening for trisomies 21, 18, and 13: Clinical application and accuracy evaluation

**Xiangmei Sun**[1,2☸], **Yuyi Ying**[1,2☸], **Caiping Chen**[1,2,3], **Xiaoliang Shi**[1,2,3]*

1 Shaoxing Maternity and Child Health Care Hospital, Shaoxing, China, 2 Maternity and Child Health Care Affiliated Hospital, Shaoxing University, Shaoxing, China, 3 Department of Prenatal Diagnosis Medical Center, Shaoxing Maternity and Child Health Care Hospital, Shaoxing, China

☸ These authors contributed equally to this work.
* 1183268992@qq.com

## Abstract

### Background

Cell-free DNA (cf-DNA) screening for common trisomies has been increasingly assimilated into prenatal care. In this study, we evaluated the Z-score accuracy and the effectiveness of non-invasive prenatal testing (NIPT) for trisomies 21, 18, and 13 using cf-DNA, and further analyze pregnancy outcomes of NIPT-positive pregnant women.

### Methods

This retrospective study analyzed pregnancies that yielded positive NIPT results at Shaoxing Maternity and Child Health Care Hospital between 01/01/2017 and 31/12/2022. Invasive prenatal diagnosis (IPD) confirmed the positive NIPT findings. Logistic regression analysis was applied to correlation analysis of Z-scores and positive predictive value (PPV). Accuracy of Z-score was evaluated through receiver operating characteristic (ROC) curve analysis. Data regarding basic characteristics, prenatal diagnosis results, and pregnancy outcomes were collected.

### Results

In total,257 high-risk cases of trisomy 21, 18, and 13 were identified. Among these,193 pregnancies underwent invasive prenatal diagnosis (IPD) in our institution, the PPV was 75.44% for T21,45.28% for T18, and 17.86% for T13. A significant association between Z-scores and PPVs were revealed by logistic regression(p < 0.05). The ROC curve analysis revealed optimal cutoff values of 7.231 for T21, 5.245 for T18, and 7.504 for T13. The corresponding areas under the curve (AUC) were 0.954, 0.941, and 0.924, respectively. Moreover, the PPV was statistically higher in the very-high-risk (VHR) group than in the general high-risk (GHR) group. In terms of pregnancy outcomes,98.51% (132/134) of pregnancies identified with chromosomal

**Data availability statement:** All relevant data for this study are available from the Figshare repository (https://doi.org/10.6084/m9.figshare.29839811.v1).

**Funding:** The author(s) received no specific funding for this work.

**Competing interests:** The authors have declared that no competing interests exist.

abnormalities were terminated, whereas 98.89% (89/90) of those diagnosed with false-positive results were carried to term.

## Conclusions

The study demonstrates that the Z-score is valuable in accurately assessing NIPT results. Consequently, clinicians can provide more efficient prenatal genetic counseling by utilizing a specific reference value for the Z-score.

## Introduction

Fetal trisomy, including trisomy 21 (T21), trisomy 13 (T13), and trisomy 18 (T18), is a developmental disorder that results in congenital disability. With no proven treatment available, comprehensive prenatal screening and diagnosis remains the only viable option for timely termination of pregnancy [1]. The field of prenatal screening was revolutionized in 1997 by the groundbreaking discovery that small fragments of extra-cellular DNA from the developing placenta could be detected in the mother's blood [2]. Noninvasive prenatal testing (NIPT) utilizes next-generation sequencing (NGS) technology to profile cell-free DNA (cf-DNA) in the mother's plasma, providing a risk-free approach. The purpose of prenatal screening is to detect any abnormalities in fetal function or structure at the earliest possible stage, providing guidance for clinically-based decision making [1,3–5]. Compared with standard screening methods, prenatal testing with cf-DNA in the general obstetric population showed significantly fewer false positives and a higher positive predictive value for detecting T21 and T18 [6]. NGS-based NIPT was introduced into clinical practice in 2010 and broadly assimilated into prenatal care in 2013. According to the American College of Obstetricians and Gynecologists (ACOG), cf-DNA is the most specific and sensitive method for screening common fetal aneuploidies [7]. However, it is important to remember that cf-DNA is not a diagnostic test as it can produce false-positive and false-negative results [8].

In order to identify aneuploidy, the differences in the proportion of cf-DNA between the target and reference chromosomes were quantified and characterized using Z-scores [1]. Additionally, a Z-score greater than 3 or less than −3 is utilized to define a high risk of chromosomal aneuploidy [9]. Multiple studies have consistently shown a significant correlation which associates the positive predictive value (PPV) performance of NIPT for common chromosome aneuploid conditions with Z-scores [10–15]. Specifically, a higher Z-score is linked to an increased probability of true positives. Maintaining scientific rigour is crucial when interpreting these findings, given the limitations of the data. These limitations encompass small sample sizes, the utilization of diverse sequencing platforms, and variations in testing procedures and Z-value classification algorithms.

In this study, NIPT was conducted using the BGISEQ-2000 sequencing platform (BGI). We examined the accuracy of Z-scores and the clinical performance of NIPT over a period of 6 years, specifically for common chromosome aneuploidies.

Additionally, we analyzed the pregnancy outcomes of women who tested positive on NIPT. These findings can aid clinicians in delivering more efficient prenatal genetic counseling through the utilization of a specific reference value for the Z-score.

## Materials and methods

### Study subjects

This retrospective study enrolled 257 pregnancies with positive NIPT results for T21, T18, and T13 at a single prenatal diagnosis center, Shaoxing Maternity and Child Health Care Hospital, from 01/01/2017 to 31/12/2022. Data were accessed by the authors for research purposes on 01/06/2024. The inclusion criteria comprised all pregnant women who underwent NIPT and tested positive during the study period, while those with negative results were excluded. All participants received thorough genetic counseling from a qualified clinician, who provided them with information regarding the test's objectives, precision, and constraints. Prior to the procedure, they voluntarily signed informed consent. The study was obtained approval from the Institutional Ethics Committee of Shaoxing Maternity and Child Health Care Hospital (Approval No. 2023047). Fig 1 presents the flowchart of the study.

### Noninvasive prenatal testing

The NIPT protocols, which involved the extraction of cfDNA, construction of libraries, and high-throughput sequencing, were performed according to the guidelines provided by the manufacturer. Maternal peripheral blood, measuring five milliliters, was collected using EDTA anticoagulant tubes. To separate the plasma layer from the whole blood, centrifugation was performed twice within a 6-hour period. Subsequently, cfDNA was extracted and enriched using a DNA extraction kit (BGI, Wuhan, China) with magnetic bead extraction on the MGISP-960 system. The DNA libraries were constructed utilizing the Fetal Chromosome Aneuploidy (T21/T18/T13) Detection Kit (BGI, Wuhan, China). The construction process

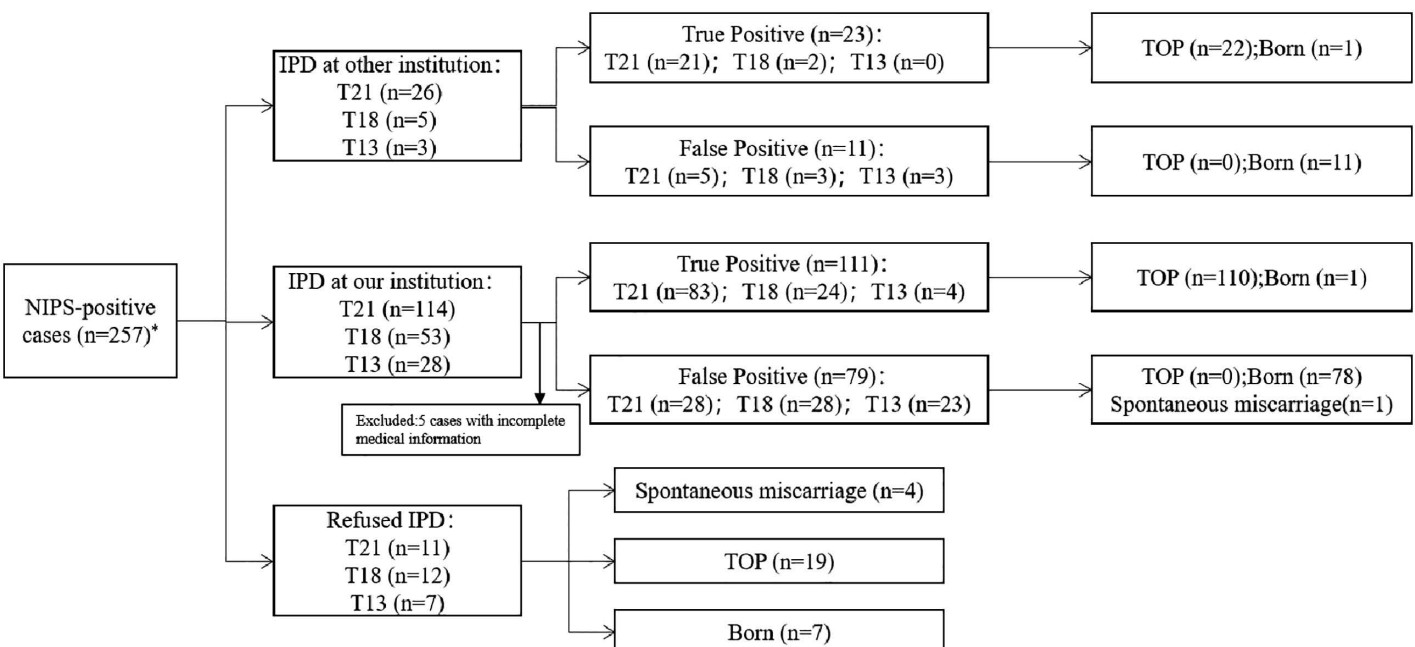

**Fig 1. Clinical outcome of all NIPS-positive cases.** Abbreviations: IPD, invasive prenatal diagnosis; TOP, termination of pregnancy; Note: *, Two cases exhibited abnormalities on two chromosomes, resulting in a total count in the box exceeding 257 by two.

included end-repair, adaptor ligation, PCR, and purification steps. Library quantification was performed using LIFE TECHNOLOGY QUBIT4.0 and the concentration greater than 2 ng/μL was considered as the qualifying standard. Subsequently,48 DNA libraries were pooled in a proportional manner to create a mixed library. After pooling, the double-stranded DNA was thermally denatured into single strands, and then created DNA circles. Qualified DNA circles were utilized in the preparation of DNA Nanoballs (DNBs) via rolling circle replication. The DNBs were assembled using a sequencing reaction kit (BGI, Wuhan, China), and then loaded onto the chip. Finally, high-throughput sequencing was conducted on the BGISEQ-2000 platform (BGI) using a combinatorial probe-anchor synthesis (cPAS) approach. The fastq data obtained from MGISEQ-2000 sequencing was aligned to the reference sequence map of the human genome (hg19). Low-quality, non-aligned, repeatedly aligned, and mismatched sequences were filtered out, and only the UniqueReads (UR) were kept. Each chromosome was subdivided into windows of a specific length. UR and GC content were calculated for each window. Z-scores were utilized to assess the likelihood of fetal chromosomal aneuploidy abnormalities. For interpreting NIPT results, Z-scores followed a normal distribution, with values beyond ±3 (|Z-score|>3) considered positive NIPT results indicative of high-risk aneuploidy, while scores within −3–3 were classified as negative results (low-risk).

### Invasive prenatal diagnosis

Pregnant individuals who received positive NIPT results were asked to return for genetic counseling. Ultrasound-guided amniocentesis or umbilical blood sampling was performed on a voluntary basis with informed consent. Clinicians and pregnant women chose one or a combination of prenatal diagnosis techniques, including karyotype analysis (Giemsa banding), chromosome microarray analysis (CMA) (CytoScanTM 750K, Affymetrix, USA), or Bacterial Artificial Chromosomes (BACs)-on-Beads assay. The procedures and analyses were conducted following applicable guidelines. The detailed technical procedures were previously reported [16].

### Follow-up of pregnancy outcomes

We utilized telephone interviews and electronic medical record systems to monitor the pregnancy outcomes of women who tested positive for NIPT.

### Statistical analysis

Statistical analyses were performed using Statistical Product and Service Solusions (SPSS) software version 26 (IBM Corp., Armonk, NY, USA). We reported the mean ± standard deviation for maternal age and gestational age at the time of testing. The result of a NIPT for an individual woman is expressed as a Z-values, where the individual sample is compared with a control group of normal (diploid) samples. In the case of an aneuploidy of a chromosome, a relative excess or deficit for that chromosome is present compared to the normal diploid situation. Z-values were expressed as median,25th percentile, and 75th percentile (P25-P75) and the analysis was performed using nonparametric tests. The PPV was calculated as the ratio of true positive cases to the total number of cases with a positive NIPT result, multiplied by 100. We utilized logistic regression analysis to investigate the associations between Z-scores and the PPV, and plotted ROC curves using Graph-Pad Prism software to identify the optimal cut-off value for predicting fetal chromosomal abnormalities. The area under the curve (AUC) quantified the test's capacity to differentiate fetal chromosomal abnormalities from normal ones. Group differences were evaluated using either the chi-square test or Fisher's exact test, with a significance level of P < 0.05.

## Results

### Basic characteristics

A total of 257 pregnancies tested positive for T21, T18, or T13 by NIPT during the 6-year study period. Among these, 193 cases underwent invasive prenatal diagnosis (IPD) at our institution, yielding positive predictive values (PPVs) of 75.44%

for T21, 45.28% for T18, and 17.86% for T13. After excluding five cases with incomplete data, 188 pregnancies were included in the final analysis: 111 with T21, 52 with T18, and 27 with T13. Table 1 presents the characteristics and clinical indications of the 188 cases. The maternal age was 32.49±5.13 years (range: 19–45 years), and the gestational age (GA) at NIPT was 16.99±3.30 weeks (range: 11+3–27+2 weeks). Moreover, the two leading indications for NIPT detection, with the highest positive predictive values (PPVs), were NT thickening (93.75%, 15/16) and ultrasound structural or soft marker abnormality (100%, 7/7).

## Analysis of logistics regression and ROC curve

As shown in Table 2, the logistic regression analyses indicated that there was a statistically significant correlation between the Z-scores and true-positive results(OR = 1.739, P<0.001). Specifically, The odds ratios for T21, T18, and T13 were 1.961(95% CI:1.498–2.566, P<0.001),2.091(95% CI:1.402–3.117, P<0.001), and 1.560 (95% CI: 1.085–2.242, P=0.016), respectively. Subsequent ROC curve analyses indicated that the optimal cutoff Z-scores for T21, T18, and T13 were 7.231, 5.245, and 7.504, respectively. The corresponding area AUC were 0.954,0.941, and 0.924 (Fig 2). Furthermore, the optimal cutoff values resulted in a sensitivity of 92.86% and a specificity of 96.39% for T21. Similarly, for T18, the sensitivity was 89.29% and the specificity was 95.83%. However, possibly due to the sample size, T13 showed a lower sensitivity of 73.91% and a higher specificity of 100%.

**Table 1. Patient characteristics and clinical indication.**

| Characteristic | Total | T21 | T18 | T13 |
|---|---|---|---|---|
| Maternal age (years) (mean±SD) | 32.49±5.13 | 32.77±5.12 | 32.27±5.08 | 32.07±5.37 |
| Gestational at NIPT test(weeks) (mean±SD) | 16.99±3.30 | 16.81±3.49 | 17.18±2.98 | 17.27±3.10 |
| Indications [TP(n)/All(n)] | | | | |
| Advanced maternal age (≥35 years) | 37/55 | 26/29 | 10/18 | 1/10 |
| Intermediate risk for maternal serum screening | 22/46 | 18/27 | 4/15 | 0/4 |
| High risk for maternal serum screening | 7/10 | 5/7 | 2/2 | 0/1 |
| Nuchal translucency thickening | 15/16 | 12/13 | 3/3 | 0/0 |
| Ultrasound structural or soft marker abnormality | 7/7 | 5/0 | 2/0 | 0/0 |
| Assisted reproductive conception | 4/8 | 2/3 | 1/4 | 1/1 |
| Adverse reproductive history | 2/3 | 1/1 | 1/1 | 0/1 |
| Voluntary demand | 11/36 | 9/20 | 0/6 | 2/10 |
| Mixed indications[a] | 6/7 | 5/6 | 1/1 | 0/0 |
| Numbers | 188[b] | 111 | 52 | 27 |

[a]Mixed indications included 2 cases of Advanced maternal age and Assisted reproductive conception,2 cases of Advanced maternal age and Nuchal translucency thickening,2 cases of Advanced maternal age and Ultrasound structural or soft marker abnormality, and 1 case of Advanced maternal age and Adverse reproductive history.

[b]There were two cases where two chromosomal anomalies occurred simultaneously, resulting in a sum in the box that is two more than 190.

**Table 2. Results of statistical analysis.**

| NIPT positve | Logistic regression ananlysis | | | | | ROC curve analysis | | | |
|---|---|---|---|---|---|---|---|---|---|
| | B | SE | Wald | OR(95%CI) | P | AUC(95% CI) | Optimal cutoff value | Sensitivity(%) | Specificity(%) |
| T21 | 0.673 | 0.137 | 24.088 | 1.961(1.498-2.566) | <0.001 | 0.9544(0.8993-1.0000) | 7.231 | 92.86 | 96.39 |
| T18 | 0.738 | 0.204 | 13.094 | 2.091(1.402-3.117) | <0.001 | 0.9405(0.8776-1.0000) | 5.245 | 89.29 | 95.83 |
| T13 | 0.455 | 0.185 | 5.766 | 1.56(1.085-2.242) | 0.016 | 0.9239(0.7974-1.0000) | 7.504 | 73.91 | 100 |
| Total | 0.553 | 0.08 | 48.048 | 1.739(1.487-2.033) | <0.001 | – | – | – | – |

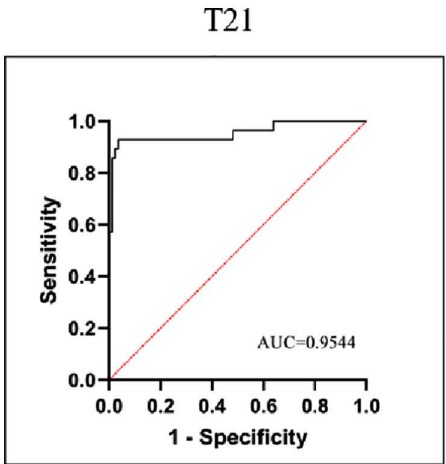
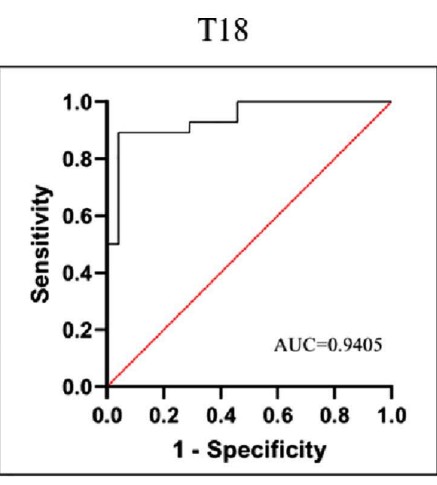
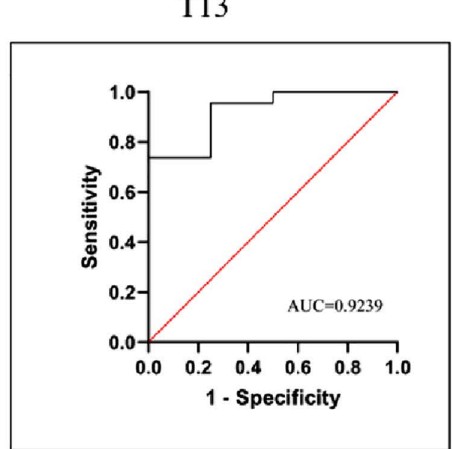

**Fig 2. The receiver operating characteristic curve analysis of noninvasive prenatal screening-positive cases.** Abbreviation: AUC, area under curve.

### Relevent literature

In Table 3, we identified six studies that evaluated the effectiveness of NIPT through ROC curve analysis. The number of cases in the studies varied from 90 to 296. All articles reported the AUC for detecting T21 (ranging from 0.89 to 0.962) and T18 (ranging from 0.80 to 0.93), with the optimal cutoff values ranging from 5.79 to 9.26 and 4.994 to 10.849, respectively. Five studies reported the optimal cutoff values for T13 (ranging from 6.889 to 12.368), while four studies reported the AUC values (ranging from 0.706 to 0.77).

### Accuracy assessment of the Z-value

As listed in Table 4, there was a variation in the distribution of Z-scores between accurate (true positive, TP) and incorrect (false positive, FP) outcomes in NIPT (T21:12.86 vs. 3.75, $P < 0.001$; T18: 9.98 vs. 3.16, $P < 0.001$; T13: 15.67 vs. 5.71, $P = 0.004$). Subsequently, pregnant individuals with positive T21/T18/T13 results were categorized into three risk groups based on the optimal cutoff values of Z-scores (the optimal cutoff Z-scores for T21, T18, and T13 were 7.231, 5.245, and 7.504, respectively): the general high-risk (GHR) group ($3 \leq Z\text{-score} <$ optimal cutoff values), the very high-risk (VHR) group (Z-score $\geq$ optimal cutoff values), and the all high-risk (AHR) group (Z-score $\geq 3$). The VHR group had significantly higher PPV compared to the GHR group for T21 (10.34% vs. 97.56, $P < 0.001$), T18 (3.85% vs. 88.46%, $P < 0.001$), and T13 (0% vs. 40%, $P < 0.001$). Additionally, there were significant differences in PPV between the GHR/VHR group and the AHR group for both T21 and T18.

Table 5 demonstrates the further categorization of Z-scores into four groups: $3 \leq Z < P25$, $P25 \leq Z < P50$, $P50 \leq Z < P75$, and $Z \geq P75$. Subsequently, the positive predictive values (PPVs) were calculated for each group separately.

### Clinical outcome of all NIPT-positive cases

As shown in Fig 1, all pregnancies were successfully followed up. Records indicate that out of the 257 IPD cases, 134 were verified as true positive. Observations from all NIPT-positive cases are as follows: pregnancies of fetuses diagnosed with common trisomies were terminated, except for one T13 case with a low level of mosaicism diagnosed at our institution, and one case of T21 with unknown reasons diagnosed at another institution. Among the 90 cases determined to be false positives, one case resulted in a spontaneous miscarriage, while the remaining 89 mothers gave birth to healthy

**Table 3. Studies reported on assessing the NIPT efficiency using ROC curve analysis.**

| Study | Sequencing platform | T21 | | | | | T18 | | | | | T13 | | | | |
|---|---|---|---|---|---|---|---|---|---|---|---|---|---|---|---|---|
| | | Sample number | Cut off value | AUC | Sensitivity (%) | Specificity (%) | Sample number | Cut off value | AUC | Sensitivity (%) | Specificity (%) | Sample number | Cut off value | AUC | Sensitivity (%) | Specificity (%) |
| Zhou et al.,2021 | Illumina Next-Seq CN500 | 120 | 5.79 | 0.89 | 93.4 | 78.67 | 38 | 6.05 | 0.8 | 80 | 76.92 | – | – | – | – | – |
| Wen et al., 2022 | BioelectronSeq 4000 | 203 | 8.997 | 0.89 | 81.18 | 97.88 | 51 | 8.1 | 0.93 | 83.78 | 92.86 | 42 | 7.409 | 0.77 | 100 | 50 |
| Wang et al.,2023 | BioelectronSeq 4000 | 59 | 9.261 | 0.949 | – | – | 14 | 10.849 | 0.875 | – | – | 17 | 12.368 | 0.75 | – | – |
| Yang et al.,2023 | BioelectronSeq 4000 | 204 | 7.597 | 0.891 | 87.94 | 80 | 52 | 4.944 | 0.928 | 96.97 | 94.44 | 40 | 9.135 | 0.796 | 97.5 | 74.07 |
| Junhui et al.,2021 | BGISEQ-500 | 135 | 6.612 | 0.954 | 96.8 | 90 | 54 | 7.574 | 0.916 | 88.9 | 92.6 | 25 | 6.889 | 0.706 | 85.7 | 61.1 |
| Chen et al.,2022 | Illumina Next-Seq CN500 and NextSeq 550Dx | 174 | – | 0.962 | – | – | 48 | – | 0.804 | – | – | 21 | – | 0.744 | – | – |
| Our study | BGISEQ-2000 | 111 | 7.231 | 0.954 | 92.86 | 96.39 | 52 | 5.245 | 0.941 | 89.29 | 95.83 | 27 | 7.504 | 0.924 | 73.91 | 100 |

**Table 4. Distribution of Z-scores and PPV performances based on the regrouping of Z-scores according to the optimal cutoff values.**

| Aneuploidy | Median (P25,P75) of Z-scores | | GHR group | | VHR group | | AHR group | |
|---|---|---|---|---|---|---|---|---|
| | True positive | False Positive | n | PPV(95% CI) | n | PPV(95% CI) | n | PPV(95% CI) |
| T21 | 12.86(10.62-19.26) | 3.75(3.33–5.20)[a] | 29 | 10.34%(2.19-27.35%) | 82 | 97.56%[b](91.47–99.70%) | 111 | 74.77%[c,d]((65.65–82.54%) |
| T18 | 9.98(7.29-15.74) | 3.16(3.00–4.40)[a] | 26 | 3.85%(0.97-19.64%) | 26 | 88.46%[b](69.85–97.55%) | 52 | 46.15%[c,d](32.23–60.53%) |
| T13 | 15.67(9.80-19.48) | 5.71(4.12–8.40)[a] | 17 | 0(0/17) | 10 | 40.00%[b](12.16–73.76%) | 27 | 14.81%(4.19-33.73%) |
| Total | – | – | 72 | 5.56%(1.53-13.62%) | 118 | 90.68%(83.93-95.25%) | 190 | 58.42%(51.06-65.51%) |

PPV, positive predictive value; GHR, general high risk group: 3≤Z-score ≤ cutoff; VHR, very high risk group: Z-score ≥ cutoff; AHR, all high risk group: Z-score ≥ 3.

[a]Stand for TP vs. FP, P<0.05;

[b]Stand for GHR group vs. VHR group, P<0.05;

[c]Stand for GHR vs. AHR group, P<0.05;

[d]Stand for VHR group vs. AHR group, P<0.05.

**Table 5. Performances of the PPV based on Z-score regrouping according to quartiles.**

| Z-score | T21 | | | T18 | | | T13 | | |
|---|---|---|---|---|---|---|---|---|---|
| | TP(n) | FP(n) | PPV(95% CI) | TP(n) | FP(n) | PPV(95% CI) | TP(n) | FP(n) | PPV(95% CI) |
| 3≤Z<P25 | 2 | 25 | 7.41%(0.91-24.29%) | 0 | 12 | 0 | 0 | 6 | 0 |
| P25≤Z<P50 | 27 | 1 | 96.43%(81.65-99.91%) | 1 | 13 | 7.14%(0.18-33.87%) | 0 | 7 | 0 |
| P50≤Z<P75 | 26 | 2 | 92.86%(76.50-99.12%) | 10 | 3 | 76.92%(46.19-94.96%) | 1 | 6 | 14.29%(0.36-57.87%) |
| Z≥P75 | 28 | 0 | 100%(87.66-100.00%) | 13 | 0 | 100%(75.29-100.00%) | 3 | 4 | 42.86%(9.90-81.59%) |

infants. Among the 30 unconfirmed cases, which account for 11.67% of all NIPT-positive cases, four patients experienced spontaneous abortion, nineteen patients underwent direct termination of their pregnancies, and live births were recorded in seven cases (including two high-risk T21 cases, two high-risk T18 cases, and three high-risk T13 cases).

## Discussion

In the past decade, prenatal screening using cf-DNA sequencing has revolutionized obstetric practice by greatly reducing the need for invasive genetic diagnostic procedures, like amniocentesis [17]. Prenatal cfDNA screening for common autosomal trisomies has several advantages, including high sensitivity, specificity, and positive predictive value. Additionally, it is a noninvasive procedure, making it the preferred choice for pregnant women [2,18,19]. The ACMG strongly recommends NIPT for all pregnant women to identify fetal trisomies 21, 18, and 13 [20].

Given the extensive application of NIPT for common trisomies, it is imperative to evaluate the Z-score accuracy and clinical performance of NIPT when interpreting individual NIPT results during counseling. When considering patient counseling, it is of greater significance to present the PPV, which represents the probability of a positive result being a TP [21]. This study found that the calculated PPV values were 74.77% for T21, 46.15% for T18, and 14.81% for T13. Prior studies indicated that the PPV range for T21 was 65.24–92.59%, the PPV range for T18 was 35.71–72.5%, and the PPV range for T13 was 10–28% [10,12,13,22–24]. In a nationwide Chinese multicenter study encompassing nearly 2 million pregnancies, noninvasive prenatal testing (NIPT) demonstrated positive predictive values (PPVs) of 69.77% for T21, 47.24% for T18, and 22.36% for T13. The analysis revealed 26 false-negative cases, corresponding to a false-negative rate of 1.4 per 100,000 pregnancies [25]. In a large-scale retrospective cohort study of 282,911 pregnancies screened NIPT, the test demonstrated robust performance for common aneuploidies, with PPVs of 86.81% for T21, 56.81% for T18, and 18.18% for T13. The false-negative rate was 2.47 per 100,000 pregnancies, representing 7 undetected cases of trisomy [26]. The findings highlight that despite NIPT's robust performance in detecting common aneuploidies, NIPT-negative pregnancies

still require ultrasound follow-up and genetic counseling, with invasive diagnostic testing recommended if abnormalities are detected. Additionally, we performed Z-score grouping based on optimal cutoff values and quartiles, as well as logistic regression and ROC curves analysis, to identify associations between Z scores and PPV. We found a meaningful association between Z-scores and true positive results in NIPT-positive cases, which is consistent with previously reported data by other researchers [12]. The AUC for common trisomies in this study consistently exceeded 0.9. Grouping Z-scores based on the optimal cutoff value revealed that the PPV of the VHR group was higher than that of the GHR group ($p<0.05$). Additionally, Z-scores were grouped into quartiles, We observed that when the Z-score exceeds the 75th percentile, the PPV for T21 and T18 reaches 100%. These findings indicate that utilizing Z-scores as a judgment indicator in NIPT yields a good diagnostic predictive value.

When evaluating clinical utilization of NIPT, it is crucial to acknowledge and not overlook the significance of both false positive and false negative (FN) outcomes. Genetic counseling should comprehensively address these findings. One potential cause of false positive results is confined placental mosaicism (CPM), which has an incidence rate of approximately 2%. CPM can be triggered either by mitotic nondisjunction events or aneuploidy rescue [27–29]. Besides CPM, false positive outcomes may also occur due to maternal malignancy, maternal chimerism, or vanishing twin syndrome [30,31]. Various factors, including maternal body mass index (BMI) and the presence of multiple pregnancies, can influence the fetal fraction to different extents, potentially resulting in false-negative outcomes [1].

The tracking of pregnancy outcomes revealed that 98.51% (132/134) of pregnancies identified with chromosomal abnormalities were terminated, while 98.89% (89/90) of false-positive results were carried to term. Pregnant with positive NIPT results are advised against quickly adopting a negative attitude. Instead, they should approach subsequent consultations with a positive mindset to identify and confirm any abnormalities through IPD. Only then can they make informed decisions about whether to continue or terminate the pregnancy [23].

However, our study had several limitations. For instance, we did not include fetal fraction, maternal copy number variation, and maternal age, all of which could potentially impact the positive rate and PPVs. Therefore, the combination of fetal fraction and other factors may offer enhanced screening capabilities, warranting further research. Conducting multicenter collaborative studies using the same technology platform is essential, which can lead to more reliable conclusions. Moreover, systematic data collection from diagnostic testing and comprehensive gathering of pregnancy outcome information are crucial for improving the accuracy of PPVs and enhancing patient counseling services.

## Conclusions

In conclusion, our study supports the notion that the Z-score is valuable in accurately assessing NIPT results. Accordingly, clinicians can enhance the efficiency of prenatal genetic counseling by utilizing a specific Z-score reference value. Importantly, NIPT cannot replace traditional diagnostic techniques, and it is prudent to rely on IPD results for prenatal decision-making.

## Author contributions

**Conceptualization:** Xiangmei Sun, Caiping Chen, Xiaoliang Shi.

**Data curation:** Yuyi Ying, Caiping Chen.

**Formal analysis:** Xiangmei Sun.

**Methodology:** Yuyi Ying.

**Validation:** Xiangmei Sun.

**Writing – original draft:** Xiangmei Sun, Yuyi Ying.

**Writing – review & editing:** Xiaoliang Shi.

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
