## [Decision Letter · Decision Letter 0]

2 Jul 2025

PONE-D-25-27034Cell-free DNA screening for trisomies 21, 18, and 13: clinical application and accuracy evaluationPLOS ONE

Dear Dr. Shi,

Thank you for submitting your manuscript to PLOS ONE. After careful consideration, we feel that it has merit but does not fully meet PLOS ONE’s publication criteria as it currently stands. Therefore, we invite you to submit a revised version of the manuscript that addresses the points raised during the review process.

We managed to get 2 reviewers to review and they have suggested some ways to improve the manuscrpt

Please see and consider resubmitting if possible

We look forward to receiving your revised manuscript.

Kind regards,

Yee Gary Ang, MBBS MPH

Academic Editor

PLOS ONE

2. For studies involving third-party data, we encourage authors to share any data specific to their analyses that they can legally distribute. PLOS recognizes, however, that authors may be using third-party data they do not have the rights to share. When third-party data cannot be publicly shared, authors must provide all information necessary for interested researchers to apply to gain access to the data. (https://journals.plos.org/plosone/s/data-availability#loc-acceptable-data-access-restrictions)

Additional Editor Comments (if provided):

Reviewers' comments:

Reviewer's Responses to Questions

**Comments to the Author**

1. Is the manuscript technically sound, and do the data support the conclusions?

Reviewer #1: Yes

Reviewer #2: Yes

2. Has the statistical analysis been performed appropriately and rigorously? 

Reviewer #1: No

Reviewer #2: Yes

3. Have the authors made all data underlying the findings in their manuscript fully available?

Reviewer #1: Yes

Reviewer #2: No

4. Is the manuscript presented in an intelligible fashion and written in standard English?

Reviewer #1: No

Reviewer #2: Yes

5. Review Comments to the Author

Reviewer #1: Methodological Clarity

• Details of the NIPS method (BGISEQ-2000) and validation (IPD) are well-described. However, the inclusion and exclusion criteria (e.g., what defines a "high-risk" group beyond Z-score) should be clarified.

o The manuscript lacks a clear description of the control population used for generating Z-scores, which is central to their interpretation.

o No analysis of fetal fraction, which is a critical confounder in NIPS.

o Limited discussion of false negatives—despite being a real concern in NIPS.

Statistical analysis

• Confidence intervals are missing from some PPV estimates and AUCs—these should be included.

Discussion: Expand comparison with other large cohort studies and meta-analyses

Referencing

- References are adequate and up-to-date, though there is inconsistent formatting.

Language: Use professional English editing service before resubmission.

Reviewer #2: Xiangmei Sun and collaborators report how z-score value could be a good predictor for NIPT results and how it correlate with PPV at high score, mainly for T21 and T18. This data is well known and this study help to corroborate this association highlighting its utility.

The paper is well written and statistical analysis are well conducted. Some minor amendements are required before publication:

- please choose if using NIPS or NIPT, maybe NIPT is more widely recognized

- in table 3 the values recorded in this study should be filled together with the previous publications in order to have a better view of the comparison among studies

- Data availability statement is missing, it is not clear where the data may be found

- please check the punctuation that is often misplaced

All the limitations of this study are well explained in the discussion section

6. PLOS authors have the option to publish the peer review history of their article (what does this mean? ). If published, this will include your full peer review and any attached files.

**Do you want your identity to be public for this peer review?** For information about this choice, including consent withdrawal, please see our Privacy Policy .Reviewer #1: No

Reviewer #2: No

---

## [Author Response · Author response to Decision Letter 1]

6 Aug 2025

We would like to thank both reviewers for their valuable comments. We feel this has helped us to improve our manuscript substantially. We have revised the manuscript accordingly, with main revisions highlighted. We also revised the manuscript according to the formatting requirements of your journal of PLOS ONE.

The detailed response to each comment is listed below in red.

Reviewer #1:

1)“Details of the NIPS method (BGISEQ-2000) and validation (IPD) are well-described. However, the inclusion and exclusion criteria (e.g., what defines a "high-risk" group beyond Z-score) should be clarified.”

Response:We sincerely appreciate the reviewer's valuable comment regarding the clarification of inclusion/exclusion criteria. In response, we have explicitly stated in the “Methods:Study subjects” section that "The inclusion criteria comprised all pregnant women who underwent NIPT and tested positive during the study period, while those with negative results were excluded."This revision provides clear study population parameters while maintaining consistency with our reported results.

2)“The manuscript lacks a clear description of the control population used for generating Z-scores, which is central to their interpretation.”

Response:As suggested,we have added this in the new manuscript.More details were in the “Methods: Noninvasive prenatal Testing” section.

3)“No analysis of fetal fraction, which is a critical confounder in NIPS.”

Response:We appreciate the reviewer’s insightful comment. We fully agree that fetal fraction (FF) is a critical confounder in NIPS analysis, and its inclusion would strengthen our study. However, due to the high proportion of missing FF data in our cohort, we intentionally excluded it from the primary analysis to avoid statistical bias.

To indirectly address this limitation, we stratified the NIPS testing gestational age into three groups (�16周, 17–20, and >20 weeks), as FF is known to correlate positively with gestational age[1-3]. No significant intergroup differences in PPV were observed. In future studies, we plan to expand the sample size and incorporate additional covariates (e.g., FF, maternal CNVs) for a more comprehensive analysis.

References

1.Gamisch A, Hess J, Mustafa-Korninger ME. Diurnal Effects on the Fraction of Fetal Cell-Free DNA in Maternal Plasma.Prenat Diagn. 2025;45(8):979-87. https://doi.org/10.1002/pd.6836 PMID: 40533246.

2.Deng C, Liu J, Liu S, Liu H, Bai T, Jing X, et al. Maternal and fetal factors influencing fetal fraction: A retrospective analysis of 153,306 pregnant women undergoing noninvasive prenatal screening. Front Pediatr. 2023;11:1066178. https://doi.org/10.3389/fped.2023.1066178 PMID: 37114008.

3.Mousavi S, Shokri Z, Bastani P, Ghojazadeh M, Riahifar S, Nateghian H. Factors affecting low fetal fraction in fetal screening with cell-free DNA in pregnant women: a systematic review and meta-analysis. BMC Pregnancy Childbirth. 2022;22(1):918. https://doi.org/10.1186/s12884-022-05224-7 PMID: 36482322.

4)“Limited discussion of false negatives—despite being a real concern in NIPS.Expand comparison with other large cohort studies and meta-analyses.”

Response:We thank the reviewer for this important point. Additional discussion on false negatives and comparisons with other studies have been added (Line 257-269).

“In a nationwide Chinese multicenter study encompassing nearly 2 million pregnancies, noninvasive prenatal testing (NIPT) demonstrated positive predictive values (PPVs) of 69.77% for T21, 47.24% for T18, and 22.36% for T13. The analysis revealed 26 false-negative cases, corresponding to a false-negative rate of 1.4 per 100,000 pregnancies[25].In a large-scale retrospective cohort study of 282,911 pregnancies screened NIPT, the test demonstrated robust performance for common aneuploidies, with PPVs of 86.81% for T21, 56.81% for T18, and 18.18% for T13. The false-negative rate was 2.47 per 100,000 pregnancies, representing 7 undetected cases of trisomy[26].The findings highlight that despite NIPT's robust performance in detecting common aneuploidies, NIPT-negative pregnancies still require ultrasound follow-up and genetic counseling, with invasive diagnostic testing recommended if abnormalities are detected.”

5)“Confidence intervals are missing from some PPV estimates and AUCs—these should be included.”

Response:As suggested, we have added confidence intervals for the PPV estimates and AUCs.

6)“References are adequate and up-to-date, though there is inconsistent formatting.”

Response:We appreciate the reviewer's feedback. We have now standardized all references according to the journal's formatting guidelines.

7)“Use professional English editing service before resubmission.”

Response:Thank you for the suggestion! We have proofread the manuscript and tried to correct grammatical errors and rephrase certain sentences.

Reviewer #2:

1)“please choose if using NIPS or NIPT, maybe NIPT is more widely recognized.”

Response:We appreciate this constructive comment. As recommended, we have uniformly adopted 'NIPT' throughout the revised manuscript.

2)“in table 3 the values recorded in this study should be filled together with the previous publications in order to have a better view of the comparison among studies.”

Response:Thank you for the suggestion! As recommended, we have integrated the data from our current study with relevant published findings in Table 3 to enable direct comparison across studies.

3)“Data availability statement is missing, it is not clear where the data may be found.”

Response:As suggested, we have added Data availability statement in the revised manuscript.The datasets for this study can be found in the this link:

https://figshare.com/s/d3fd982f9b830a52ec4f.

4)“please check the punctuation that is often misplaced.”

Response: We have proofread the manuscript as suggested.

We tried our best to improve the manuscript and made some changes highlighted with different colors in revised paper. We appreciate for Editors/Reviewers’ warm work earnestly, and hope the correction will earn your recognition and approval. Once again, thank you very much for your comments and suggestions.

---

## [Decision Letter · Decision Letter 1]

26 Aug 2025

Cell-free DNA screening for trisomies 21, 18, and 13: clinical application and accuracy evaluation

PONE-D-25-27034R1

Dear Dr. Shi,

We’re pleased to inform you that your manuscript has been judged scientifically suitable for publication and will be formally accepted for publication once it meets all outstanding technical requirements.

Kind regards,

Yee Gary Ang, MBBS MPH

Academic Editor

PLOS ONE

Additional Editor Comments (optional):

Reviewers' comments:

Reviewer's Responses to Questions

**Comments to the Author**

1. If the authors have adequately addressed your comments raised in a previous round of review and you feel that this manuscript is now acceptable for publication, you may indicate that here to bypass the “Comments to the Author” section, enter your conflict of interest statement in the “Confidential to Editor” section, and submit your "Accept" recommendation.

Reviewer #1: (No Response)

Reviewer #3: All comments have been addressed

2. Is the manuscript technically sound, and do the data support the conclusions?

Reviewer #1: Yes

Reviewer #3: Yes

3. Has the statistical analysis been performed appropriately and rigorously? 

Reviewer #1: Yes

Reviewer #3: Yes

4. Have the authors made all data underlying the findings in their manuscript fully available?

Reviewer #1: Yes

Reviewer #3: Yes

5. Is the manuscript presented in an intelligible fashion and written in standard English?

Reviewer #1: Yes

Reviewer #3: Yes

6. Review Comments to the Author

Reviewer #1: Thank you for your revisions. The manuscript is now clearer, more coherent, and easier to understand. The study’s limitations have also been acknowledged to guide further research.

Reviewer #3: The authors have successfully addressed all concerns, and the manuscript has been revised accordingly.

7. PLOS authors have the option to publish the peer review history of their article (what does this mean? ). If published, this will include your full peer review and any attached files.

**Do you want your identity to be public for this peer review?** For information about this choice, including consent withdrawal, please see our Privacy Policy .

Reviewer #1: No

Reviewer #3: No

---

## [Editor Report · Acceptance letter]

PONE-D-25-27034R1

PLOS ONE

Dear Dr. Shi,

I'm pleased to inform you that your manuscript has been deemed suitable for publication in PLOS ONE. Congratulations! Your manuscript is now being handed over to our production team.

Kind regards,

on behalf of

Dr. Yee Gary Ang

Academic Editor

PLOS ONE